# Pulmonary and Extrapulmonary Manifestations of Fungal Infections Misdiagnosed as Tuberculosis: The Need for Prompt Diagnosis and Management

**DOI:** 10.3390/jof8050460

**Published:** 2022-04-28

**Authors:** Bassey E. Ekeng, Adeyinka A. Davies, Iriagbonse I. Osaigbovo, Adilia Warris, Rita O. Oladele, David W. Denning

**Affiliations:** 1Medical Mycology Society of Nigeria, Lagos 101017, Nigeria; dynkodunsi1@gmail.com (A.A.D.); iyabo.osaigbovo@uniben.edu (I.I.O.); oladelerita@gmail.com (R.O.O.); 2Department of Medical Microbiology and Parasitology, University of Calabar Teaching Hospital, Calabar 540271, Nigeria; 3Infectious Diseases Unit, Department of Internal Medicine, University of Calabar, Calabar 540271, Nigeria; 4Department of Medical Microbiology and Parasitology, Olabisi Onabanjo University Teaching Hospital, Sagamu 121102, Nigeria; 5Department of Medical Microbiology, School of Medicine, College of Medical Sciences, University of Benin, Benin City 300213, Nigeria; 6Medical Research Council Centre for Medical Mycology, University of Exeter, Exeter EX4 4QD, UK; a.warris@exeter.ac.uk; 7Department of Medical Microbiology and Parasitology, Faculty of Basic Medical Sciences, College of Medicine, University of Lagos, Lagos 101017, Nigeria; 8Manchester Fungal Infection Group, Faculty of Biology, Medicine and Health, University of Manchester, Manchester Academic Health Science Centre, Manchester M13 9WL, UK; ddenning@manchester.ac.uk

**Keywords:** fungal, infections, tuberculosis, diagnosis, pulmonary

## Abstract

Fungal infections commonly present with myriad symptoms that mimic other clinical entities, notable amongst which is tuberculosis. Besides histoplasmosis and chronic pulmonary aspergillosis, which can mimic TB, this review has identified several other fungal infections which also do. A total of 80 individual cases misdiagnosed as TB are highlighted: aspergillosis (*n* = 18, 22.5%), histoplasmosis (*n* = 16, 20%), blastomycosis (*n* = 14, 17.5%), cryptococcosis (*n* = 11, 13.8%), talaromycosis (*n* = 7, 8.8%), coccidioidomycosis (*n* = 5, 6.3%), mucormycosis (*n* = 4, 5%), sporotrichosis (*n* = 3, 3.8%), phaeohyphomycosis (*n* = 1, 1.3%) and chromoblastomycosis (*n* = 1, 1.3%). Case series from India and Pakistan reported over 100 cases of chronic and allergic bronchopulmonary aspergillosis had received anti-TB therapy before the correct diagnosis was made. Forty-five cases (56.3%) had favorable outcomes, and 25 (33.8%) died, outcome was unclear in the remainder. Seventeen (21.3%) cases were infected with human immunodeficiency virus (HIV). Diagnostic modalities were histopathology (*n* = 46, 57.5%), culture (*n* = 42, 52.5%), serology (*n* = 18, 22.5%), cytology (*n* = 2, 2.5%), gene sequencing (*n* = 5, 6.3%) and microscopy (*n* = 10, 12.5%) including Gram stain, India ink preparation, bone marrow smear and KOH mount. We conclude that the above fungal infections should always be considered or ruled out whenever a patient presents with symptoms suggestive of tuberculosis which is unconfirmed thereby reducing prolonged hospital stay and mortalities associated with a delayed or incorrect diagnosis of fungal infections.

## 1. Introduction

Tuberculosis (TB) is the 13th leading cause of death globally and a foremost infectious killer. It is estimated that 10 million people developed TB worldwide with about 1.5 million deaths in 2020 alone (WHO, 2020). Fungal diseases affect over a billion people with about >1.6 million deaths annually, yet the index of suspicion for fungal infections amongst clinicians remains poor compared to TB [1]. This is particularly concerning for high TB burdened countries in sub-Saharan Africa where a significant proportion of the population including HIV/AIDS patients are at risk of fungal infections [2]. Similarities in the patterns of presentation of invasive fungal infections and TB results in misdiagnosis often associated with increased length of hospital stay, economic loss, increased morbidity and poor clinical outcomes [2]. Patients in areas endemic for tuberculosis including Africa and Asia are often commenced on anti-TB therapy despite negative microbiological test results including the highly sensitive nucleic acid amplification tests [2,3]. In addition, the true burden of fungal infections especially in resource limited settings has been masked by lack of diagnostic tools giving the impression that they are uncommon compared to well-known conditions such as TB [2]. The challenge of false-positive TB diagnoses due to pressure from National TB program to reduce the incidence and deaths from TB, has also contributed to the above narrative [4]. This review highlights case reports initially managed as TB based on signs and symptoms, and radiological findings but later confirmed to be fungal infections. The aim is to emphasize the necessity of considering fungal infections as possible differential diagnoses at the outset in patients with signs and symptoms suggestive of TB, thus invariably improving clinical outcomes.

## 2. Search Criteria

A systematic literature search was conducted using PubMed, Google Scholar, AJOL, Cochrane Library and grey literature to identify case reports and case series on fungal infections mimicking or misdiagnosed as TB between 1 January 1960 and 31 December 2021. The following search terms were used: “aspergillosis and tuberculosis”, “*Aspergillus* and tuberculosis”, “histoplasmosis and tuberculosis”, “*Histoplasma* and tuberculosis”, “mucormycosis and tuberculosis”, “chromoblastomycosis and tuberculosis”, “sporotrichosis and tuberculosis”, “paracoccidioidomycosis and tuberculosis”, “coccidioidomycosis and tuberculosis”, “phaeohyphomycosis and tuberculosis”, and/or “invasive fungal infections and tuberculosis”. References in all relevant papers were also reviewed for additional publications (‘snow balling’) on case reports regarding the topic that may not have been published in the searched databases. Publications without patients’ country of origin were excluded. Only publications of original case reports written in English were included. Data extracted from each case report included the following: gender, age, clinical features, investigation/diagnostic measures, treatment and outcomes. Case reports with defined HIV status were documented as such, while case reports with undefined HIV status were assumed to be negative.

## 3. Results and Discussion

Our extensive literature search revealed 80 cases of fungal infections misdiagnosed as TB. Forty (50%) were from Asia with 23 (23/40, 57.5%) from India, 26 (32.5%) from Africa, 8 (10%) from North America, 5 (6.3%) from Europe and 1 (1.3%) from South America. The fungal infections identified were aspergillosis (*n* = 18, 22.5%), histoplasmosis (*n* = 16, 20%), blastomycosis (*n* = 14, 17.5%), cryptococcosis (*n* = 11, 13.8%), talaromycosis (*n* = 7, 8.8%), coccidioidomycosis (*n* = 5, 6.3%), mucormycosis (*n* = 4, 5%), sporotrichosis (*n* = 3, 3.8%), phaeohyphomycosis (*n* = 1, 1.3%) and chromoblastomycosis (*n* = 1, 1.3%). Seventeen (21.3%) cases were infected with human immunodeficiency virus (HIV). Diagnostic modalities were histopathology (*n* = 46, 57.5%), culture (*n* = 42, 52.5%), serology (*n* = 18, 22.5%), cytology (*n* = 2, 2.5%), gene sequencing (*n* = 5, 6.3%) and microscopy (*n* = 10, 12.5%) including Gram stain, India ink preparation, bone marrow smear and KOH mount. Underlying morbidities/risk factors were only documented in 11 case reports. Disseminated fungal infections including forms affecting the CNS, vertebra and skin accounted for 52.5% (*n* = 42) of cases while pulmonary fungal infections were 47.5% (*n* = 38). Forty-five cases (56.3%) had favorable outcomes, 25 (31.3%) died, the outcome was not stated in 9 (11.3%) and 1 (1.3%) lost to follow up. A statistically significant relationship was observed when comparing the relationship between fatal outcomes and HIV status of patients (*p*
*<* 0.05, Fisher’s exact test). The clinical summary of cases is highlighted in Table 1.

### 3.1. Aspergillosis

The ubiquitous fungus *Aspergillus* causes aspergillosis, an opportunistic infection with a global distribution. Infection occurs due to failure of clearance of *Aspergillus* spores from the lungs, which then germinate and grow, especially within damaged lungs or existing lung cavities [5]. Pulmonary *Aspergillus* infection also directly leads to cavity formation. Common antecedents include previous lung infection including any *Mycobacterium*, sarcoidosis, chronic obstructive pulmonary disease (COPD) or allergic bronchopulmonary aspergillosis (ABPA) [70,71]. A study from Vietnam identified 38 cases of chronic pulmonary aspergillosis (CPA), out of which ten were previously treated for recurrent TB once, and three treated for TB three times [72]. Another study from India reported that at least 38 of 100 patients with CPA post-TB had been re-treated for TB [73]. Chronic pulmonary aspergillosis (CPA) is often misdiagnosed as smear/gene Xpert negative TB [5,6,8,9,10], especially in TB-endemic countries, because patients present with symptoms and/or chest imaging similar to TB [10,11]. A cross sectional study from Brazil identified 15 (7%) cases of pulmonary mycoses in 213 patients managed as smear negative TB. Of the fifteen, ten were aspergillosis, three were paracoccidioidomycosis and one case each of histoplasmosis and cryptococcosis [74]. Another study from Nigeria, reported a CPA prevalence of 8.7% in 208 smear negative TB patients [75]. ABPA can also be misdiagnosed as TB. A retrospective study from Pakistan reported 71 patients with ABPA out of which 63 (88.7%) were being managed as a case of smear negative TB and had received empiric anti-TB therapy once (*n* = 52, 82.5%), twice (*n* = 8, 12.6%) or thrice (*n* = 3, 4.7%) [76]. This finding was similar to that of a case report from Thailand in a 19-year-old with ABPA who had received anti-TB treatment twice [19].

The spectrum of CPA ranges from chronic cavitary pulmonary aspergillosis (CCPA) which can progress to chronic fibrosing pulmonary aspergillosis (CFPA) when left untreated, single aspergilloma and sub-acute invasive (chronic necrotizing) pulmonary aspergillosis (SAIA) [70,77]. Aspergilloma characteristically develops in a pre-existing lung cavity [78]. It is a late complication of CPA, non-invasive and mostly affects an upper lobe with air crescent signs on imaging [79]. It may be multiple or single and may exist alone or with allergic bronchopulmonary aspergillosis (ABPA) [71]. SAIA affects mildly immunocompromised individuals and has symptoms and radiological features similar to CCPA, but progresses over weeks, not months [8]. Radiologically, CCPA presents with one or more cavities, usually with pleural thickening and para-cavitary infiltrates [70]. These cavities may be single or multiple and surrounded by thin or thick wall. Other less frequent radiological findings are solid nodules, solid mass or combination of both with ill-defined or lobulated margins [70,75]. The most common symptom of patients with CCPA is cough, while life-threatening hemoptysis is reported in approximately 12% to 43% [79]. Other characteristic symptoms are persistent chest pain or discomfort, weight loss and fatigue; the presence of fever may indicate SAIA. Night or day sweats are occasionally reported [80]. Out of the 18 cases, 17 were HIV negative, 9 were treated with anti-tuberculosis therapy despite negative AFB/GeneXpert, skin tuberculin or culture result with one treated based on chest imaging without AFB/GeneXpert being carried out. The outcome of one case was not stated, 12 had favorable outcomes, while in 5 the outcome was death (Table 1 and Table 2).

### 3.2. Histoplasmosis

Histoplasmosis occurs worldwide, is endemic in the United States of America and Latin America; and is commonly associated with HIV/AIDS in the adult population [80]. In the pediatric population, on the other hand, histoplasmosis is predominantly associated with risk factors other than HIV including environmental exposures and toxins, autoimmune diseases, childhood malignancies as well as their treatment, lung diseases, immunosuppressive therapies, pancytopenia, T-cell deficiency and malnutrition [80,81]. *Histoplasma capsulatum* var *capsulatum* is associated with infections in humans, while the variety *farciminosum* causes infection in horses. Eight clades (North American class 1 clade, North American class 2 clade, Latin American group A clade, Latin American group B clade, Australian clade, Netherlands clade, Eurasian clade and African clade) have been identified within *Histoplasma capslatum* with five additional phylogenetic species within Latin America (LAm A1, LAm A2, LAm B1, LAm B2, RJ and BAC-1) [82,83]. The fungus is primarily found in soil enriched with bird or bat guano [84]. Human infection primarily occurs by inhalation of microconidia, followed by the development of pulmonary disease which may become disseminated in immunocompromised individuals [84]. Despite being highlighted in several reviews as a fungal disease mimicking TB, histoplasmosis is still misdiagnosed as TB especially in Sub Saharan Africa, South East Asia and Latin America, where the availability and accessibility to diagnostic tools and antifungal therapy remains a major challenge (Table 2) [2,20,21,22,23,24,25,26,27]. Cases of histoplasmosis and TB coinfections are also reported [2,85]. Poor clinical outcomes are associated with a delay in diagnosis or the inability to make a prompt diagnosis of dual TB and histoplasmosis and poor accessibility to appropriate antifungal drugs [2,21,23,25,26]. We summarized 16 case patients with histoplasmosis misdiagnosed as TB; 10 were HIV positive, 8 had negative AFB/Gene Xpert results, AFB/Gene Xpert results were not stated in 8 cases, 9 had dismal outcomes, 2 of which were diagnosed at autopsy (Table 1 and Table 3).

### 3.3. Blastomycosis

Blastomycosis is caused by the dimorphic fungus *Blastomyces dermatitidis*. Other recently discovered species include *Blastomyces percursus*, *Blastomyces emzantsi* and *Blastomyces gilchristii*. It is endemic in the region that is adjacent to lakes and rivers such as the Mississippi river, North America, Southern Canada and Africa. Activities such as the disruption of soil and vegetation increase the risk for infection [86,87]. Inhalation of the spores results in asymptomatic, acute, chronic or disseminated infection [86,87,88]. *Blastomyces dermatitidis* is a primary pathogen and can cause disease in immunocompetent people [86]. The clinical presentation and radiological features are similar to those caused by other chronic lung diseases and may cause diagnosis to be delayed or missed [28,29]. The radiographic findings are non-specific, they include consolidation, pulmonary infiltrates, cavitation, solid lobulated mass, and solitary, spiculated nodules as seen in TB or malignancy, Table 4. Of the 14 case reports, all were HIV negative and all were treated for tuberculosis despite negative mycobacterial culture and AFB/geneXpert. Three died due to delayed diagnosis and commencement of antifungal therapy, while nine recovered (the outcome in two were not reported) (Table 1 and Table 4).

### 3.4. Cryptococcosis

Cryptococcosis is an opportunistic infection caused by pathogenic encapsulated yeasts in the genus *Cryptococcus* with more than 30 species ubiquitously distributed in the environment, and most abundant in the droppings of pigeons and other birds [89]. *C. neoformans* and *C. gattii* are the two species commonly known to cause human disease. *C. neoformans* causes disease in both immunocompromised and immunocompetent hosts, while *C. gattii* is commonly associated with infections in immunocompetent patients [89]. Cryptococcal infection predominantly occurs in immunocompromised individuals including advanced HIV disease (AHD) patients, patients with leukemia, lymphoma, and diabetes mellitus, with AHD accounting for more than 80% of cases globally [47,90]. Infection occurs primarily by inhalation of basidiospores from environmental reservoirs with deposition into pulmonary alveoli [89]. Most cryptococcal infections are either pulmonary or cerebromeningeal [42]. The cerebromeningeal forms predominantly occur in the AHD population but has been reported in the immnocompetent [47]. Disseminated forms affecting other body sites including the skin, prostate, eyes, and bone/joints are less common [89]. The clinical features of pulmonary cryptococcosis may mimic pulmonary TB resulting in misdiagnosis of cases especially in areas highly endemic for TB including Asia and Africa [39,42,43]. In addition, abnormal radiologic findings in cryptococcal disease including nodules, pulmonary infiltrates, pleural effusions, hilar lymphadenopathy, lung cavitation and osteolytic lesions are also found in TB [39,40,41,42,43,44,45,46,47,48,49]. Furthermore, in the absence of India ink, the large encapsulated yeast may be mistaken for lymphocytes in CSF microscopy. Cryptococcal infections may also coexist with TB in severely immunocompromised individuals [91]. Delayed diagnosis often results in death [40,41,43,46,47]. Of the 11 patients with cryptococcosis misdiagnosed as TB, 9 were negative for AFB/Gene Xpert, results of AFB/Gene Xpert were not stated in two cases, three were HIV positive, five died due to delayed diagnosis and commencement of antifungal therapy and six recovered (Table 1 and Table 5).

### 3.5. Talaromycosis

Talaromycosis is caused by the dimorphic fungus *Talaromyces marneffei* endemic in South East Asia, East Asia and South Asia [92,93,94]. It is a major cause of HIV-associated opportunistic infections in endemic regions, making up to 16% of hospital admissions due to AIDS [93,94]. An increasing number of talaromycosis cases have been reported in HIV-uninfected patients [95]. Human infection results from inhalation of fungal spores following disruption of the soil [94]. The clinical features are non-specific and are indistinguishable from those of disseminated tuberculosis and other systemic mycoses, with disseminated infection involving multiple organ systems as the most common manifestation in patients with advanced HIV disease [94,96]. In addition, *T. marneffei* infection of the respiratory system is often misdiagnosed as TB, with fatal complications. A retrospective study from China reported 63 patients with *T. marneffei* respiratory system infection of which 24 (38.1%) were misdiagnosed as tuberculosis. Seven of the patients did not receive antifungal therapy and died from severe systemic inflammatory responses whereas of the 56 patients who received antifungal therapy, 15 died during the first round of antifungal therapy due to worsened clinical conditions and organ failure [97]. Cases of *T. marneffei* and *Mycobacterium tuberculosis* coinfection have also been reported [98]. We summarize seven cases of talaromycosis earlier managed as TB cases, only one case was HIV positive, AFB/Gene Xpert findings were not stated in four cases but were negative in three, the outcome was favorable in all except one, Table 1 and Table 6.

### 3.6. Coccidioidomycosis

*Coccidioides* species, namely *C. imitis* and *C. posadasii*, are dimorphic fungi that are found as mold in soil. It is endemic in southwestern United States, Central America, and South America with *C. immitis* largely confined to California, whereas *C. posadasii* predominates in Arizona, Texas, South America and part of Mexico [99]. People with extensive exposure to soil or dusty environments such as farmers, construction workers, or those that perform outdoor activities such as hunting, and soil digging are at risk of *Coccidioides* infection [99]. The infectious unit is a single arthroconidia, and symptoms can range from asymptomatic to acute pneumonia, often with cutaneous immunological reactions such as erythema nodosum. Chronic pneumonia and/or disseminated disease may follow [100]. About 1% of infected persons develop disseminated disease [56,57,60], which can involve the skin, joints, bones, central nervous system, or other organs and factors such as HIV, organ transplantation, diabetics mellitus, immunosuppressive agents, advanced age, pregnancy and Filipino and African ethnicity has an increased risk [99,101]. Chest imaging features may reveal cavities, opacities, pleural effusion and miliary nodules similar to chest radiological findings in TB thereby causing misdiagnosis of cases as shown in Table 6, especially if out of an endemic area. Of the five patients, all were HIV negative, two had underlying morbidity, none were AFB/GeneXpert positive, four were commenced on anti-TB regimen despite having negative AFB/GeneXpert results, all had favorable outcomes with treatment (Table 1 and Table 6).

### 3.7. Mucormycosis

Mucormycosis is a destructive fungal infection commonly encountered in immunocompromised patients, especially in diabetics, patients with hematological malignancies, recipients of allogeneic hematopoietic stem cell transplants, immunosuppressive agents and chemotherapy [62]. It consists of a group of opportunistic infections caused by fungi in the class Zygomycetes and order Mucorales [62]. The genera commonly associated with infections in humans are *Rhizopus*, *Mucor* and *Absidia* [62,64]. They are ubiquitous and predominantly found in the soil or decaying organic matter. Infection usually occurs via the inhalation of spores with pulmonary forms occasionally mimicking the clinical presentation of pulmonary TB and may be misdiagnosed as such [61,62,63,64]. Usually, pulmonary mucormycosis is an acute or subacute infection, with a poor outcome, but occasionally it can be a chronic disease process. In addition, case reports on concomitant mucormycosis and TB, and cases of TB complicated by mucormycosis have been reported [102,103]. Mucormycosis should therefore be considered in patients with suspected TB, especially when there is no improvement with anti-tuberculous therapy in patients with risk factors outlined above. Mortality often occurs with delayed presentation and/or diagnosis [63]. Besides fatalities, economic losses and prolonged hospital stay are other challenges identified in the highlighted case reports and these were not unconnected with poor awareness and low index of suspicion on the part of clinicians [61,62,63,64]. We summarized four cases of pulmonary mucormycosis misdiagnosed as pulmonary tuberculosis; AFB/GeneXpert results were negative in two case patients and not stated in the other two cases, only one case was HIV positive, outcome was favorable in two cases, not revealed in one, while the other had fatal outcome (Table 1 and Table 6).

### 3.8. Sporotrichosis

*Sporothrix species* are thermally dimorphic fungi found in trees, rose bushes, and grasses. In South America a separate species (*S. brasiliensis*) causes feline sporotrichosis and consequently human infection. Acute or sub-acute granulomatous infection follows traumatic skin inoculation with conidia [67]. It may manifest as a cutaneous, pulmonary or disseminated disease and is considered an occupational risk for gardeners, farmers, horticulturists and forest workers [67]. Males are commonly affected [65,66,67], and 75% of cases are lymphocutaneous, but in immunocompromised people, particularly those with HIV, disseminated sporotrichosis has been reported [104,105]. Primary pulmonary sporotrichosis occurs when the conidia of the *Sporothrix species* are inhaled and present as a chronic cavitary fibronodular disease that usually affects middle-aged people with chronic alcoholism or chronic obstructive pulmonary disease [65]. The clinical and radiological findings are non-specific and as such could be mistaken for other chronic pulmonary disorders, such as TB [65,66,67]. Sporotrichosis is common in Brazil, Peru, South Africa, India, Japan and China [106,107]. Outbreaks caused by zoonotic transmission from cats have been reported in Rio de Janeiro, Brazil, Argentina, Paraguay, Panama and Uruguay [106,108,109]. Cat-to-cat and/or cat-to-human transmissions through bites, scratches or contact with cutaneous exudative lesions from an infected cat spread the disease [110,111]. Additionally, zoonotic transmission has been reported from other animals such as armadillo, rats, horses, and squirrel [110,111]. Of the three cases mistaken as TB, one had contact with an animal, one was a farmer and all three had pulmonary sporotrichosis. Two improved with antifungal therapy but one died due to delay in diagnosis and treatment (Table 1 and Table 6).

### 3.9. Phaeohyphomycosis

Phaeohyphomycosis is a spectrum of clinical syndromes that includes allergic disease, keratitis, cutaneous and subcutaneous disease, pulmonary infections, infections of the central nervous system and disseminated disease [112]. It is caused by pigmented filamentous fungi that are commonly described as dematiaceous or melanized molds [110]. Some of the genera implicated in human infections include *Curvularia*, *Alternaria*, *Exophiala*, *Phialophora*, *Lomentospora*, *Verruconis*, *Chaetomium* and *Rhinocladiella.* They exist in the environment as saprophytes and phytopathogenic agents [68]. A particular feature of these fungi is that they rarely cause infections in immunocompetent individuals [68]. This was illustrated by a case of cerebral phaeohyphomycosis mimicking TB reported by Hesarur et al., in which the patient had no features suggestive of immunocompromise: HIV serology was negative, and not on any chronic medications known to suppress immunity. The only risk factor was farming (Table 1 and Table 6) [68]. The fungal pathogen, *Fonsecaea pedrosoi* isolated in this case is found in rotten wood, decaying plant materials, and soil with a worldwide distribution, particularly in tropical Asia, South America, and Africa [68]. Hence, farming as postulated by the authors must have exposed the patient to this infection. Besides mimicking TB, phaeohyphomycosis has also been reported to complicate TB [113,114].

### 3.10. Chromoblastomycosis

Chromoblastomycosis is an indolent cutaneous infection caused by several dematiaceous fungi. It is usually found in tropical and subtropical areas [69,115]. There are diverse genera, these are *Fonsecaea pedrosoi*, *Phialophora verrucosa*, *Fonsecaea compactum*, *Cladophialophora carrionii*, *Exophiala jeanselmei*, *Exophiala castellanii* and *Rhinocladiella aquaspersa* but the most prevalent causal agent is *Fonsecaea pedrosoi.* It can take the form of plaque-like, nodular, verrucous or cicatricial lesions [69,115]. The risk increases with occupation, and the inoculation of the skin with soil or vegetable matter contaminated with these pigmented fungi causes disease [69,115]. It forms a localized lesion at the site of inoculation that can rarely spread through the lymphatic or blood. Its presentation may mimic cutaneous tuberculosis [69,115], squamous cell carcinoma, sporotrichosis and psoriasis in appearance [69]. A high index of suspicion especially in areas endemic for TB and a histological examination of biopsied tissue is invaluable in making a definitive diagnosis (Table 1 and Table 6).

**Table 6 jof-08-00460-t006:** Cases of talaromycosis, coccidioidomycosis, mucormycosis, sporotrichosis, chromoblastomycosis and phaeohyphomycosis misdiagnosed as TB.

Authors	Sex/Age/ Location	AFB/Gene Xpert	HIV Status	Investigations	Treatment	Outcome
**Talaromycosis**						
Chen Q et al., 2021 [50]	F/43/China	NS	-	Metagenomic next-generation sequencing, Culture	Voriconazole	Fv
Chen Q et al., 2021 [50]	M/49/China	NS	-	Metagenomic next-generation sequencing, Culture	LAMB and Voriconazole, Itraconazole.	Fv
Wang et al., 2020 [51]	W/33/China	NS	-	Next-generation sequencing	Voriconazole	Fv
Lee PP et al., 2019 [52]	M/9/China	NS	-	Histopathology	AMB, Flucytosine and fluconazole	Fv
Cheng-yan et al., 2021 [53]	M/5/China	-	-	Microscopy, Culture	AMB and Voriconazole	D
Sethuraman et al., 2020 [54]	M/37/ India	-	+	Culture	Itraconazole	Fv
Fan et.al., 2021 [55]	M/2/China	-	-	Culture, Biopsy	Voriconazole, AMB, Itraconazole	Fv
**Coccidioidomycosis**						
China et al., 2014 [56]	F/15/USA	-	-	Culture	Fluconazole	Fv
Kshitij et al., 2016 [57]	M/47/India	-	-	Histopathology, Culture	Itraconazole	Fv
Peralaya et al., 2021 [58]	M/57/India	-	-	Culture	Itraconazole	Fv
Chauhan et al., 2019 [59]	M/62/India	-	-	Histopathology	AMB	L
Capoor et al., 2014 [60]	M/31/India	NS	-	Fine needle aspiration, Culture	AMB and Itraconazole	Fv
**Mucormycosis**						
Wen-Fang et al., 2010 [61]	M/51/China	NS	-	Histopathology	LAMB and flucytosine	Fv
Divya et al., 2021 [62]	M/26/India	-	+	Microscopy	AMB	NS
Kim et al., 2020 [63]	M/32/Korea	NS	-	Histopathology	LAMB	D
Garg et al., 2008 [64]	M/70/India	-	-	Culture	AMB	Fv
**Sporotrichosis**						
Orofino-Costa et al., 2013 [65]	M/32/Brazil	NS	-	Serology, Histopathology, Culture, Gene sequences	AMB, Itraconazole	Fv
Singhai et al., 2012 [66]	M/22/India	-	-	Microscopy, Culture	Itraconazole	Fv
Kinas et al., 1976 [67]	M/48/Jamaica	NS	-	Culture, Serology	AMB	D
**Phaeohyphomycosis**						
Hesarur et al., 2020 [68]	M/48/India	NS	-	Histopathology, Culture	AMB and Voriconazole.	Fv
**Chromoblastomycosis**						
Arghya et al., 2015 [69]	F/50/India	NS	-	Histopathology, Culture	Itraconazole	Fv

M; Male, F; Female, AFB; Acid fast bacilli, HIV; Human immunodeficiency virus, LAMB; Liposomal Amphotericin B, AMB; Amphotericin B, -; Negative, +; Positive, NS; Not stated, Fv; Favorable, D; Death, L; Lost to follow up.

## 4. Conclusions

There is a need for more awareness on fungal infections mimicking TB and their clinical presentations especially in countries endemic for tuberculosis. Making a diagnosis of TB and commencement of anti-TB therapy despite negative AFB/GeneXpert results in suspected TB patients without the consideration of possible fungal infections reiterates the gaps that need to be urgently addressed. The fatalities documented in the case reports highlighted in this review were due to delayed diagnosis or misdiagnosis of fungal infections in patients presumed to have TB. Most of the cases were from Asia and Africa, in particular India and Nigeria, which are countries with a high burden of TB, and low resource countries with less facilities for diagnosis and poor accessibility to appropriate antifungal therapy. Paradoxically, few cases of fungal infections misdiagnosed as TB were found in South America.

Results from a survey of laboratory practices for diagnosis of fungal infection carried out in seven Asian countries emphasized the need for the development of quality laboratories, accreditation and training of manpower in existing laboratories and accessibility to advanced non-culture-based diagnostic tests such as galactomannan, beta D glucan and PCR [116]. In South America, the results of a survey involving 129 centers in 24 countries showed a lack of diagnostic tools and availability of therapy, with one quarter of the institutions surveyed reported to have no access to ‘cryptococcal antigen tests’ and a significant percentage (39%) of institutions having no access to antifungal susceptibility tests [117]. In a gap analysis survey from 22 tertiary hospitals spread across the six geopolitical zones of Nigeria, Osaigbovo et al. identified the absence of a mycology laboratory in 22.7% (5/22), no access to a biosafety cabinet in 22.7% (5/22), lack of laboratory scientists trained in mycology in 40.9% (9/22), lack of participation in external quality assurance in 100% and no antifungal sensitivity testing in 77.3% (17/22) of institutions. Galactomannan, cryptococcal antigen lateral flow assay and latex agglutination tests were used in 4.5% (1/22), 13.6% (3/22) and 22.7% (5/22) of institutions, respectively [118]. In another survey on the current state of clinical mycology in Africa conducted across 21 African countries, access to susceptibility testing for both yeasts and molds was available in only 30% of institutions. *Aspergillus* spp. antigen detection was available in only 47.5% of institutions as an in-house or outsourced test, while access to mold-active antifungal drugs such as amphotericin B deoxycholate, voriconazole and posaconazole were available in only 52.5%, 35.0% and 5.0% of institutions, respectively [119].

Prompt diagnosis will improve the outcome of fungal infections. This can be achieved by sending appropriate specimens including bronchoalveolar lavage, bone marrow aspirate, sputum and blood for fungal culture [2,6,23,28,41,54,58,64] especially in the absence of any microbiological evidence for TB. Persistence of symptoms in confirmed TB patients on anti-TB therapy also necessitates investigation for fungal infections as co-infections can occur. Definitive diagnosis can also be made from tissue biopsy for histopathology [2,5,29,39,52,59,65,68,69]. Direct microscopy using India ink and Giemsa staining can identify *Cryptococcus* spp. and *Histoplasma* spp., respectively [2,44,46,47]. Screening for fungal antigens and immunoglobulins including *Histoplasma* antigen, cryptococcal antigen, *Aspergillus* galactomannan, *Aspergillus* IgG and IgE are effective tools for establishing a diagnosis of disseminated histoplasmosis, cryptococcosis, invasive aspergillosis, CPA and ABPA, respectively [2,8,18,19,20,41,42]. Culture, together with clinical features, is the most favored diagnostic option for sporotrichosis, direct microscopy and histopathology is challenging due to the frequent absence of microorganisms detectable by these methods [120]. Direct microscopy, histopathology and culture with clinical features is recommended for the diagnosis of chromoblastomycosis [120]. Molecular assays including PCR and gene sequencing are also available but not sustainable for the routine diagnosis of fungal infections, especially in resource limited settings [37,50,51,65,121].

Patients presumed to have tuberculosis should be simultaneously investigated for fungal infections as an important differential diagnosis especially in endemic areas. In addition, readily incorporating diagnostic testing for fungal disease into routine practice would greatly reduce the morbidity and mortality of fungal infections erroneously mismanaged as tuberculosis. The statistically significant relationship between fatal outcomes and HIV infection in this review further emphasizes the need to screen people living with HIV/AIDS for opportunistic infections other than TB. In a recent national program report from Guatemala, rapid screening for histoplasmosis, TB and cryptococcosis in people living with HIV was shown to decrease mortality by 7% [122]. A high index of suspicion, improved diagnostics and routine screening for fungal infections in TB endemic regions will invariably curb the rate of misdiagnosis and lead to better patient outcomes in the affected regions.

## Figures and Tables

**Table 1 jof-08-00460-t001:** Clinical summary of fungal infections misdiagnosed as TB.

Final Fungal Infection Diagnosis	Clinical Features of Fungal Infections Mimicking TB	Comorbidity/Exposure	Organ(s) Affected	HIV Status	Diagnostic Measures	Definitive Treatment Following Fungal Diagnosis	Outcomes	Ref.
Aspergillosis (*n* = 18)	Hemoptysis, cough, pleuritic chest pain, fever, wheeze, weight loss, dyspnea, paraplegia, right-sided weakness,	Farming (*n* = 1)	lung (*n* = 15) Vertebrae (*n* = 2) CNS (*n* = 1)	17 HIV- 1 HIV+	Histopathology (*n* = 6), Culture (*n* = 8), Serology (*n* = 12) Chest CT/X-ray (*n* = 7)	Surgery (*n* = 4), AMB (*n* = 2), Keto (*n* = 1), Itra (*n* = 10), Vori (*n* = 5)	Fv (*n* = 12) D (*n* = 5) NS (*n* = 1)	[5,6,7,8,9,10,11,12,13,14,15,16,17,18,19]
Histoplasmosis (*n* = 16)	Fever, generalized weakness, abdominal pain, abdominal swelling, nocturnal sweating, skin lesions, lymphadenopathy, hepatomegaly, splenomegaly	DM (*n* = 1), CLD (*n* = 1)	lung (*n* = 1), disseminated (*n* = 15)	10 HIV+ 6 HIV-	Histopathology (*n* = 15), Culture (*n* = 1), Serology (*n* = 1), Cytology (1)	LAMB (*n* = 1), AMB (*n* = 6), Flu (*n* = 4), Itra (*n* = 3), Keto (*n* = 1)	Fv (*n* = 3), D (*n* = 9), NS (*n* = 4)	[2,20,21,22,23,24,25,26,27]
Blastomycosis (*n* = 14)	Hemoptysis, loss, fatigue, cough, fever, weight loss, headache, altered mentation, skin lesions, pleuritic chest pain	-	lung (*n* = 6) Vertebrae (*n* = 7) CNS (*n* = 1)	14 HIV-	Histopathology (*n* = 8), Culture (*n* = 10) Gene sequencing (*n* = 2) Microscopy (*n* = 3)	Itra (*n* = 4), AMB (*n* = 9) Vori (*n* = 2), Flu (*n* = 2)	Fv (*n* = 9) D (*n* = 3) NS (*n* = 2)	[28,29,30,31,32,33,34,35,36,37,38]
Cryptococcosis (*n* = 11)	Fever, cough, difficulty in walking, lethargy, weight loss, night sweats, chest pain, backache and weakness of the limbs, lymphadenopathy	DM (*n* = 3) Farming (*n* = 1)	Lung (*n* = 5), disseminated (*n* = 6) Vertebrae (*n* = 2)	3 HIV+ 8 HIV-	Histopathology (*n* = 7), Culture (*n* = 8), Cytology (*n* = 1), Serology (*n* = 2), Microscopy (*n* = 3)	LAMB (*n* = 1), AMB (*n* = 9), Flu (*n* = 5), Fluc (*n* = 3)	Fv (*n* = 6), D (*n* = 5)	[39,40,41,42,43,44,45,46,47,48,49]
Talaromycosis (*n* = 7)	Fever, lymphadenopathy, cough, night sweats, swelling, headache, dizziness, blurred vision and vomiting.	-	lung (*n* = 5) disseminated (*n* = 2)	7 HIV-	Histopathology (*n* = 2), Culture (*n* = 5), Microscopy (*n* = 1) Gene sequencing (*n* = 3)	LAMB (*n* = 1), AMB (*n* = 2), Vori (*n* = 5), Itra (*n* = 2), Flu (*n* = 1), Fluc (*n* = 1)	Fv (*n* = 6) D (1)	[50,51,52,53,54,55]
Coccidioidomycosis (*n* = 5)	Cough, fever, discharging sinus, weight loss, lymphadenopathy	IHD (*n* = 1), SLE (*n* = 1)	lung (*n* = 1) disseminated (*n* = 4)	5 HIV-	Histopathology (*n* = 3), Serology (*n* = 1), Culture (*n* = 4), Microscopy (*n* = 1)	Flu (*n* = 1) Itra (*n* = 3) AMB (*n* = 2)	Fv (*n* = 3) L (*n* = 1) NS (*n* = 1)	[56,57,58,59,60]
Mucormycosis (*n* = 4)	Fever, cough, chest pain, hemoptysis	DM (*n* = 1)	lung (*n* = 4)	3 HIV+ 1 HIV-	Histopathology (*n* = 2), Culture (*n* = 1), Microscopy (*n* = 1)	LAMB (*n* = 1), AMB (*n* = 3), Fluc (*n* = 1)	Fv (*n* = 2), NS (*n* = 1), D (*n* = 1)	[61,62,63,64]
Sporotrichosis (*n* = 3)	Cough, sputum fever, weight loss, weakness, breathlessness,	Contact with horses (*n* = 1)	lung (*n* = 2) disseminated (*n* = 1)	3 HIV-	Histopathology (*n* = 1), Culture (*n* = 3), Serology (*n* = 2), Microscopy (*n* = 1)	Surgery (*n* = 1), AMB (*n* = 2) Itra (*n* = 1)	Fv (*n* = 2) D (1)	[65,66,67]
Phaeohyphomycosis (*n* = 1)	Chronic headache, fever, impaired vision and hearing	--	CNS	1 HIV-	Histopathology (*n* = 1), Culture (*n* = 1)	AMB andVori	Fv	[68]
Chromoblastomycosis (*n* = 1)	Multiple hyperpigmented verrucous plaque	-	Cutaneous	1 HIV-	Histopathology (*n* = 1), Culture (*n* = 1)	Itra	Fv	[69]

*n*; Number of cases, Fv; Favorable outcome, D; Death, L, Lost to follow-up, NS, Not stated, HIV; Human immunodeficiency virus, LAMB; Liposomal Amphotericin B, AMB; Amphotericin B, Flu; Fluconazole, Itra; Itraconazole, Fluc; Flucytosine, Vori; Voriconazole, IHD; Ischaemic heart disease, DM; Diabetes mellitus, SLE; Systemic lupus erythematosus, CLD; Chronic liver disease.

**Table 2 jof-08-00460-t002:** Cases of aspergillosis misdiagnosed as TB.

Authors	Sex/Age/ Location	AFB/ Gene Xpert	HIV Status	Investigations	Treatment	Outcome
Maheshwar et al., 2011 [5]	M/50/India	-	-	Histopathology	Lobectomy	Fv
Ur-Rahman et al., 2000 [6]	F/40/Pakistan	-	-	Histopathology, Culture	Surgical decompression, AMB and Itraconazole	D
Mckee et al., 1984 [7]	M/22/Columbia	NS	-	Histopathology, Culture	Surgical decompression and. AMB	Fv
Desgranges et al., 2014 [8]	M/30/France	-	-	Chest CT, Culture, Histopathology, Serology	Voriconazole	Fv
Diengdoh et al., 1983 [9]	M/34/London	-	-	Histopathology, Culture	Ketoconazole	Fv
Kant et al., 2007 [10]	F/39/India	-	-	Chest CT, Serology	Prednisolone, Budesonide and Salbutamol	Fv
Singh et al., 2018 [11]	F/35/Indonesia	NS	-	Chest CT, Serology	Methyl- prednisolone and Itraconazole	Fv
Neki et al., 2017 [12]	F/40/India	-	-	Chest CT, Serology	Prednisolone and Itraconazole	Fv
Ahmed et al., 2020 [13].	M/56/India	-	-	Chest CT, Serology	Itraconazole	Fv
Gbajabiamila et al., 2018 [14]	M/35/Nigeria	-	-	Serology	Itraconazole	Fv
Davies et al., 1978 [15]	F/38/UK	-	-	Culture	-	NS
Davies et al., 1978 [15]	M/25/UK	NS	-	Culture, Serology	Left pneumonectomy.	D
Kwizera et al., 2021 [16]	F/45/Uganda	-	-	Chest X ray, CT-scan	Itraconazole	Fv
Kwizera et al., 2021 [16]	F/53/Uganda	NS	+	Chest X ray, Serology	Itraconazole	D
Kwizera et al., 2021 [16]	F/18/Uganda	-	-	Serology	Itraconazole, ceftriaxone and levofloxacin.	D
^a^ Green et al., 1969 [17]	M/41/New Orlean	NS	-	Histopathology	-	D
Patil et al., 2018 [18]	M/48/India	-	-	Culture, Serology	Itraconazole and omnacortil	Fv
Mizuhazhi et al., 2021 [19]	19/Thailand	-	-	Culture, Serology	Itraconazole and steroids	Fv

M; Male, F; Female, AFB; Acid fast bacilli, HIV; Human immunodeficiency virus, AMB; Amphotericin B, -; Negative, +; Positive, NS; Not stated, Fv; Favorable, D; Death, a; Diagnosis was made at autopsy.

**Table 3 jof-08-00460-t003:** Cases of histoplasmosis misdiagnosed as TB.

Authors	Sex/Age/Location	AFB/ Gene Xpert	HIV Status	Investigations	Treatment	Outcome
Ramesh et al., 2021 [20]	M/38/India	NS	-	Histopathology, Serology	LAMB and Itraconazole	Fv
^a^ Cipriano et al., 2020 [21]	M/30/Guinea Bissau	-	+	Histopathology	-	D
Qureshi. 2008 [22]	M/5/Pakistan	-	-	Histopathology	NS	NS
^b^ Tong et al., 1983 [23]	F/45/Singapore	-	-	Culture, Histopathology	-	D
Mendengue et al., 2021 [2]	F/29//Uganda	NS	+	Histopathology	AMB and fluconazole	Fv
^c^ Mendengue et al., 2021 [2]	F/34/Cameroon	NS	+	Culture, Histopathology	-	D
^c^ Mendengue et al., 2021 [2]	M/32/Ivory Coast	-	+	Histopathology	-	D
Mendengue et al., 2021 [2]	41/M/Congo	-	+	Histopathology	AMB	NS
Mendengue et al., 2021 [2]	26/F/Congo	NS	+	Histopathology	AMB and Itraconazole	D
Mendengue et al., 2021 [2]	44/M/Congo	NS	+	Histopathology	Ketoconazole, AMB and Itraconazole	D
Khalil et al., 1997 [24]	M/14/Nigeria	NS	-	Histopathology	NS	NS
Mendengue et al., 2021 [2]	M/45/S. Afr	NS	+	Histopathology	AMB and fluconazole	D
Mendengue et al., 2021 [2]	M/35/S. Afr.	NS	+	Histopathology	Fluconazole and Itraconazole	Fv
Mosam et al., 2006 [25]	F/11/S. Afr.	-	+	Histopathology	AMB	D
^b^ Kabangila et al., 2011 [26]	M/12/Tanzania	-	-	Histopathology	-	D
Pamnani et al., 2010 [27]	F/6/Kenya	-	-	Cytology	NS	NS

M; Male, F; Female, AFB; Acid fast bacilli, HIV; Human immunodeficiency virus, LAMB; Liposomal Amphotericin B, AMB; Amphotericin B, -; Negative, +; Positive, NS; Not stated, Fv; Favorable, D; Death, a; Death due to lack of antifungals, b; Diagnosis was made at autopsy, c; Death prior to definite diagnosis.

**Table 4 jof-08-00460-t004:** Cases of blastomycosis misdiagnosed as TB.

Authors	Sex/Age/Location	AFB/ Gene Xpert	HIV Status	Investigations	Treatment	Outcome
Matthew et al., 2018 [28]	M/25/USA	-	-	Culture	Itraconazole	Fv
Krumpelbeck et al., 2018 [29]	M/37/USA	NS	-	Culture, Histopathology	AMB and Voriconazole	NS
Goico et al., 2018 [30]	M/36/USA	NS	-	Culture	AMB and Itraconazole	Fv
Guler et al., 1995 [31]	M/4/Turkey	NS	-	Histopathology	Surgery and AMB	Fv
Kumar et al., 2019 [32]	M/32/India	-	-	Microscopy, Culture	Itraconazole	Fv
Frean et al., 1992 [33]	M/40/S. Afr	-	-	Microscopy, Culture	AMB	D
Frean et al., 1992 [33]	M/31/S. Afr	-	-	Histopathology, Culture	AMB	Fv
Zhao et al., 2011 [34]	M/21/China	-	-	Histopathology, Culture	AMB and Itraconazole	Fv
Koen F et al., 1999 [35]	M/42/S. Afri	-	-	Histopathology, Culture	Fluconazole & Itraconazole	Fv
Krulsselbrink et al., 2010 [36]	F/37/Canada	NS	-	Microscopy, Culture	AMB and Fluconazole	NS
Mapanga et al., 2020 [37]	M/31/S. Afri	NS	-	Histopathology, Gene sequencing	AMB	Fv
Mapanga et al., 2020 [37]	M/40/S. Afr	NS	-	Histopathology, Culture, Gene sequencing	AMB	D
Baily et al., 1991 [38]	F/38/Mozambique	NS	-	Culture	-	D
Baily et al., 1991 [38]	M/12/Zimbabwe	NS	-	Histopathology	AMB	Fv

M; Male, F; Female, AFB; Acid fast bacilli, HIV; Human immunodeficiency virus, AMB; Amphotericin B, -; Negative, +; Positive, NS; Not stated, Fv; Favorable, D; Death.

**Table 5 jof-08-00460-t005:** Cases of Cryptococcosis misdiagnosed as TB.

Authors	Sex/Age/ Location	AFB/ Gene Xpert	HIV Status	Investigations	Treatment	Outcome
Jain et al., 1999 [39]	F/72/India	-	-	Histopathology, Cytology, Culture	AMB and flucytosine	Fv
Shimoda et al., 2014 [40]	M/51/Japan	NS	+	Culture	-	D
Gupta R et al., 2012 [41]	M/30/India	-	-	Serology, Culture	AMB and fluconazole	D
Nakatudde et al., 2021 [42]	M/8/Uganda	-	-	Culture, Histopathology, Serology	AMB and fluconazole	Fv
Fasih N et al., 2015 [43]	M/55/Pakistan	-	-	Histopathology, Culture	AMB	D
Sompal et al., 2017 [44]	F/14/India	-	-	Histopathology, Microscopy	AMB and fluconazole	Fv
Adsul et al., 2019 [45]	F/45/India	NS	-	Histopathology, Culture	AMB and flucytosine	Fv
Ismail et al., 2018 [46]	F/5/India	-	-	Microscopy, Culture	AMB and Flucytosine	D
Jarvis et al., 2009 [47]	F/27/S. Afr	-	+	Microscopy, Culture	AMB	D
Joshi et al., 2020 [48]	F/38/India	-	-	Histopathology	AMB and fluconazole.	Fv
Tweddle et al., 2012 [49]	F/35/UK	-	+	Histopathology	LAMB and fluconazole	Fv

M; Male, F; Female, AFB; Acid fast bacilli, HIV; Human immunodeficiency virus, LAMB; Liposomal Amphotericin B, AMB; Amphotericin B, -; Negative, +; Positive, NS; Not stated, Fv; Favorable, D; Death.

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
