# Peer review of "Pulmonary and Extrapulmonary Manifestations of Fungal Infections Misdiagnosed as Tuberculosis: The Need for Prompt Diagnosis and Management"

_jof, 2022, doi:10.3390/jof8050460_

Round 1
Reviewer 1 Report
The review is quite comprehensive and detailed. The manuscript will require some spelling and grammar check, some of the references weren't cited properly e.g the number of the reference was not in brackets therefore can be quite confusing to the reader. E.g. lines 114 and 115.
Can the authors comment on the availability of the diagnostic tests in Asia and Africa which may contribute to the delayed diagnosis?
What prompted evaluation for fungal infections - was it the negative tests for TB? Shouldn't fungal infections be included in the differential diagnosis at the onset considering that a significant proportion of these patients are immunocompromised, to avoid delay in the diagnosis and as the author stated has led to unfavorable outcomes?
What about the role of molecular assays? Are these available as well?
I think its important to highlight the lack of important diagnostic modalities in certain areas of the world which contributes to the poor outcome of some patients.
Author Response
Thank you very much for your review. Please find our responses in the brackets by your comments
- Some of the references weren't cited properly e.g the number of the reference was not in brackets therefore can be quite confusing to the reader. E.g. lines 114 and 115. (All references now properly cited)
- Can the authors comment on the availability of the diagnostic tests in Asia and Africa which may contribute to the delayed diagnosis? (We have commented on that. Kindly refer to line 382-401)
- What prompted evaluation for fungal infections - was it the negative tests for TB? Shouldn't fungal infections be included in the differential diagnosis at the onset considering that a significant proportion of these patients are immunocompromised, to avoid delay in the diagnosis and as the author stated has led to unfavorable outcomes? (We have emphasized the need to consider fungal infections at the outset. kindly refer to line 421– 429)
- What about the role of molecular assays? Are these available as well? (This has been highlighted. kindly refer to line 418-420)
- I think it’s important to highlight the lack of important diagnostic modalities in certain areas of the world which contributes to the poor outcome of some patients. (We have done so. Kindly refer to line 382-401)
Reviewer 2 Report
This is a well-written paper that addresses a public health issue.
Introduction
I would recommend reviewing lines 44 and 45. I understand the point to compare COVID and TB deaths, but is it necessary?
Lines 53 and 54 repeat the idea of “endemic areas”
Lines 58-60 require a better explanation
Methods
The objective of this manuscript was to emphasize the necessity of considering fungal infections as possible differential diagnoses in patients with signs and symptoms suggestive of TB, thus it is also important to look for manuscript that compares the clinical manifestations of both infections. You could add some information.
Also, you mention that the review literature included manuscripts before 2022, could you clarify the period of the search?
Results
Table 1 clarify that column 1 refers to the final fungal infection diagnosis
Table 1 cites “treatment” it is the treatment after fungal infection diagnosis? Clarify
Discussion
I recommend unifying the discussion section and do not separate per aetiologic agent. The results of each table (Tables 2-6) should be transferred to the results section. It seems like a results section. Some information is already included in Table 1, such as the outcome. Also, you should combine the results of tables 2-6 in one table, and discuss some characteristics.
Table 5 had some blue text
Discuss the overall mortality of the misdiagnosed fungal infections
Conclusion
Lines 342-347, should include the necessity to perform a screening for both infections, especially in endemic areas
Line 356-370 mentions giemsa for Cryptococcus; however new methods such as the LFA test had high performance. Some authors mention that the antigen test for CrAg and histo should be the new gold standard. Clarify this idea.
You should discuss the performance of the TB test into the discussion.
Author Response
Thank you very much for your review. Please find our responses in the brackets by your comments
Introduction
- Reviewer: I would recommend reviewing lines 44 and 45. I understand the point to compare COVID and TB deaths, but is it necessary? (We have done so)
- Lines 53 and 54 repeat the idea of “endemic areas” (It’s just for emphasis. The challenge of fungal infections misdiagnosed as TB is more common in high TB burdened countries and this is reflected in this review as well)
- Lines 58-60 require a better explanation. (We have done so)
Methods
- The objective of this manuscript was to emphasize the necessity of considering fungal infections as possible differential diagnoses in patients with signs and symptoms suggestive of TB, thus it is also important to look for manuscript that compares the clinical manifestations of both infections. You could add some information (We had cited publications that compared the clinical manifestations of both infections. kindly refer to reference no 2,3,74-76).
- Also, you mention that the review literature included manuscripts before 2022, could you clarify the period of the search? (We have done so, kindly refer to line 69)
Results
- Table 1 clarify that column 1 refers to the final fungal infection diagnosis (We have done so)
- Table 1 cites “treatment” it is the treatment after fungal infection diagnosis? Clarify (We have done so)
Discussion
- I recommend unifying the discussion section and do not separate per aetiologic agent (We have considered your suggestion, but we think that formatting the discussion per etiologic agent makes the organization, clarity, and understanding and the specifics for each infection more appropriate for the reader). The results of each table (Tables 2-6) should be transferred to the results section. It seems like a results section (Transferring the result of each table to the result section is a good idea as suggested, and we have amended the manuscript accordingly with a clear distinction between results and discussion) Some information is already included in Table 1, such as the outcome (Table 1 is a summary, with individual data provided in the tables 2-6, and therefore it is not a repetition), Also, you should combine the results of tables 2-6 in one table, and discuss some characteristics (We have considered this suggestion, but this will have a negative impact on the clarity and organization of the tables)
- Table 5 had some blue text
- Discuss the overall mortality of the misdiagnosed fungal infections (we have done so. Kindly refer to lines 98-101, 425-429)
Conclusion
- Lines 342-347 should include the necessity to perform a screening for both infections, especially in endemic areas (we have done so. Kindly refer to line 367-368)
- Line 356-370 (now 408-409) mentions giemsa for Cryptococcus; however new methods such as the LFA test had high performance. (Giemsa was mentioned for Histoplasmosis not cryptococcosis). Some authors mention that the antigen test for CrAg and histo should be the new gold standard. Clarify this idea. (culture remains the gold standard but has however been used in few studies. In addition, the it takes a longer time and may not yield growth of pathogens in individuals with mild disease. Antigen tests is more suitable for the diagnosis of disseminated disease and monitoring of therapy)
- You should discuss the performance of the TB test into the discussion (this is not within the scope of this manscript).
Reviewer 3 Report
1- LINE 322. Is it correct that Fonseca pedrosoi was the fungi ?
2- In the tables the form for classifying the result of the therapy must be unified. In table 1 F,D,NS,L were used and in the other tables R,D,NS,LTF
3- LINE 153-154 Change: "Histoplasma spp associated with human infections are Histoplasma capsulatum and Histoplasma duboisii".
Phylogenetic studies have defined at least eight clades within H capsulatum: two North American, two Latin American and one each of Australian, Indonesian, Eurasian and African clades. More recent analysis has led to a proposal of new phylogenetic species nested within the former Latin American group A clade.
Phylogeography of the fungal pathogen H capsulatum . Mol Ecol 2003;12(12):3383-401. https://doi.org/10.1046/j.1365-294x.2003.01995.x.e
Worldwide phylogenetic distributions and population dynamics of the genus Histoplasma. Plos Negl Trop Dis 2016. https://doi.org/10.1371/journal.pntd.0004732
Author Response
Reviewer 3
Thank you very much for your review. Please find our responses in the brackets by your comments
- Line 322, now 350. Is it correct that Fonseca pedrosoi was the fungi? (The authors of the cited publication stated that fungal culture of dura mater obtained via biopsy yielded Fonseca pedrosoi. More details are available at Hesarur, N.; Seshagiri, D.V.; Nagappa, M.; Rao, S.; Santosh, V.; Chandrashekar, N. et. al. Case Report: Chronic Fungal Meningitis Masquerading as Tubercular Meningitis. J. Trop. Med. Hyg. 2020, 103, 1473–1479)
- In the tables the form for classifying the result of the therapy must be unified. (Classification of results across tables have now been unified)
- Line 153-154 (now line 181-186) (modified as suggested and references cited)
Round 2
Reviewer 2 Report
The changes have been addressed. I think that the manuscript provide clear information and make awareness of an important problem.
Author Response
Dear Reviewer, thank you very much for your review.
It was very helpful
Kind regards
Ekeng